# Dendritic Cells and Immunogenic Cancer Cell Death: A Combination for Improving Antitumor Immunity

**DOI:** 10.3390/pharmaceutics12030256

**Published:** 2020-03-12

**Authors:** María Julia Lamberti, Annunziata Nigro, Fátima María Mentucci, Natalia Belén Rumie Vittar, Vincenzo Casolaro, Jessica Dal Col

**Affiliations:** 1Departamento de Biología Molecular, Universidad Nacional de Río Cuarto, Río Cuarto 5800, Córdoba, Argentina; mjulialamberti@gmail.com (M.J.L.); fatimamentucci@gmail.com (F.M.M.); 2INBIAS, CONICET-UNRC, Río Cuarto 5800, Córdoba, Argentina; 3Department of Medicine, Surgery and Dentistry ‘Scuola Medica Salernitana’, University of Salerno, 84081 Baronissi, Salerno, Italy; annnigro@unisa.it (A.N.); vcasolaro@unisa.it (V.C.)

**Keywords:** immunogenic cell death, dendritic cell-based vaccination, immunotherapy, cancer treatment

## Abstract

The safety and feasibility of dendritic cell (DC)-based immunotherapies in cancer management have been well documented after more than twenty-five years of experimentation, and, by now, undeniably accepted. On the other hand, it is equally evident that DC-based vaccination as monotherapy did not achieve the clinical benefits that were predicted in a number of promising preclinical studies. The current availability of several immune modulatory and targeting approaches opens the way to many potential therapeutic combinations. In particular, the evidence that the immune-related effects that are elicited by immunogenic cell death (ICD)-inducing therapies are strictly associated with DC engagement and activation strongly support the combination of ICD-inducing and DC-based immunotherapies. In this review, we examine the data in recent studies employing tumor cells, killed through ICD induction, in the formulation of anticancer DC-based vaccines. In addition, we discuss the opportunity to combine pharmacologic or physical therapeutic approaches that can promote ICD in vivo with in situ DC vaccination.

## 1. Introduction 

Dendritic cells (DCs) represent the sentinels of the immune system and they play an important role in linking innate and adaptive immune responses [1]. Since their discovery in 1973 by Ralph Steinman (2011 Nobel Prize for Medicine or Physiology) [2], they have been proposed as the most powerful professional antigen presenting cells (APCs), given their ability to stimulate unprimed (naïve) helper and cytotoxic T cells and perform antigen cross-presentation [3].

Immature DCs (imDCs) have the ability to uptake antigens, process them, and present the antigenic peptides on surface MHC molecules [4]. The migration of these DCs, now mature DCs (mDCs), to lymphoid organs is triggered by chemokine-related signaling. Once in the lymph nodes, they stimulate naïve T cells through T cell receptor (TCR) binding, which provides the so-called “signal 1” for proper, antigen-specific T-cell activation [5]. In addition, the engagement of the CD28-CD80/CD86 and CD40-CD40L molecule pairs provides a necessary co-stimulatory signal, also referred to as “signal 2”. Finally, DCs paracrine stimuli also provide “signal 3”, which is associated to polarization and differentiation of T cells into effector cells. Pattern recognition receptors (PRRs) ligation by pathogen-associated molecular patterns (PAMPs) or by damage-associated molecular patterns (DAMPs) has a crucial role in the proper delivery of signal 2 and signal 3. These events have been designated “signal 0”, which constitutes the upstream signal that starts the immune response by directing DC maturation and migration [1,6].

In the tumor microenvironment (TME), the infiltration by mDCs has been associated with an increase in the recruitment of immune effector cells and pathways [7]. Anti-tumor CD8+ T cell activity is induced by DCs, which acquire, process, and present tumor-associated antigens (TAAs) on MHC molecules (signal 1) and provide co-stimulation (signal 2) and soluble factors (signal 3) to shape the T cell responses [8,9]. Unfortunately, these processes are often hampered in cancer patients due to TME-mediated suppression. Tumor-induced modulation of tumor-infiltrating DCs frequently leads to their dysfunction, which results in failure in signal 1, 2, and/or 3 and negative interference of anti-tumor CD8+ T cell immunity. For that reason, DC-mediated cross-presentation of tumor antigens in cancer patients often induces T cell tolerance instead of immunity [10,11,12].

In the last two decades, several studies have shown that some anticancer agents, including chemotherapeutics and physical therapeutic modalities, can exert immunomodulatory activities, which directly affect immune cells in the TME, including DCs, or, indirectly, modifying the cancer cell immunogenicity by the induction of immunogenic cell death (ICD) [13,14,15]. Most of these anticancer treatments promote ICD through reactive oxygen species (ROS) generation and Endoplasmic Reticulum (ER) stress [16,17,18], causing the release or the surface exposure of a series of DAMPs through a well-defined spatiotemporal scheme [19].

Normally, these DAMPs are molecules that are present inside live cells and exerting different physiological functions, but in conditions of cellular stress, damage, or death they are exposed or secreted, becoming able to interact with pattern recognition receptors (PRRs) that are expressed by immune cells, especially by DCs. This interaction leads to an immune response that is usually correlated with the establishment of immunological memory [16,20]. The essential involvement of DCs in the immune responses triggered when cancer cells undergo ICD has been described in different studies [18,21,22,23,24,25,26,27,28], which suggests that the ability of ICD inducers to stimulate an efficient antitumor T cell response depends on the presence and the susceptibility to the activation of DCs in the TME.

Advances in understanding the features of the immune TME have led to promising developments in cancer-therapeutic strategies that exploit DC manipulation to enhance their immune-stimulatory capacity. The proper management of DCs holds great potential for synergizing with other therapeutic approaches currently in use, which are aimed at boosting efficient antitumor immunity. The present review investigates and discusses the opportunity to combine the potentiality of DC-based immunotherapy with the peculiar anticancer activity of ICD inducers.

## 2. Dendritic Cell-Based Anticancer Immunotherapies

Although DCs are a rare tumor-infiltrating immune cell population, the central role of these cells in orchestrating tumor-specific immunity and tolerance is well documented and has been highlighted in recent clinical trials of immune checkpoint inhibitors (ICIs). Indeed, tumors that were grafted into mice deficient for conventional CD8α+/CD103+ DCs (also called cDC1) did not respond to immunotherapy with anti-PD1 and anti-PDL1 ICI [29,30]. Moreover, the infiltration of human cDC1 within human tumors is associated with responsiveness to anti-PD1 treatment [31]. Recently, Garris et al. clearly showed how the cross-talk between T cells and DCs is required for successful anti-PD1 immunotherapy [32]. In addition, the efficacy of the adoptive transfer of CD8+ T cells was also correlated with the extent of DC infiltration in patients with melanoma [33]. These clinical evidences have raised renewed interest in protocols improving DC-based immunotherapies that, in the past, resulted in limited, or only sporadic, success. Different strategies to exploit DC function for anticancer therapy have been investigated: the inoculation of DC activators within the tumor, the delivery of adjuvant/antigen carriers by nanoparticles directly targeting DCs, or the administration of ex vivo generated, loaded-, or unloaded-DCs. Each approach presents advantages and limitations that highlight that the improvement of some aspects is still needed. Herein, we focus our attention on current protocols for patient vaccination with ex vivo generated autologous DCs.

### 2.1. DC-Based Vaccines

Most of the studies aimed at exploiting antigen-specific stimulatory capacity of DCs have been based on the employment of ex vivo differentiated and antigen-loaded autologous DCs. This approach entails the development of DC differentiation protocols, the choice of the antigens for DC loading, and the proper selection of adjuvants in the vaccine formulation. Although the relationship between ex vivo generated DCs and the different subsets of natural DCs is not fully elucidated [34,35,36], the availability of convenient, established protocols to expand and differentiate DCs from patient-derived monocytes (mo-DCs) has strongly supported the employment of mo-DCs in the majority of clinical trials [3,37]. Different protocols and cytokine cocktails can be used for differentiating mo-DCs in sufficient numbers for vaccination. The combination of granulocyte-macrophage colony-stimulating factor (GM-CSF) with interleukin (IL)-4 or interferon-α (IFN-α) is the method that is most frequently used to generate mo-DCs with an immature or semi-mature phenotype [38,39]. These mo-DCs secrete high levels of IL-12p70 and prime T cells to preferentially activate Th1 type and cytotoxic T lymphocyte (CTL) responses [38]. The gold standard maturation cocktail to obtain completely mature mo-DCs includes the pro-inflammatory cytokines tumor necrosis factor (TNF)-α, IL-1β, and IL-6, as well as prostaglandin E2 (PGE2), in different combinations [40]. This approach is the most commonly used, even though its actual advantage relative to the use of GM-CSF/IL-4 alone has not been conclusively demonstrated. Moreover, it is well known that PGE2 can also exert several immunosuppressive or tolerogenic functions, despite the important effects of PGE2 on DC migration and T cell proliferation [41]. Alternative methods have been successfully tested for ex vivo maturation of DCs, including the induction of co-stimulatory (CD40-CD40L or CD137L agonists) or the activation pathways (Toll-like receptor (TLR) and PRR agonists) [3]. 

Another fundamental step in DC-based vaccine development is the mo-DCs loading with TAAs. It was shown that using multiple TAAs, instead of one or few defined antigens, could achieve significant clinical benefits and avoid the rapid evolution of tumor escape variants based on specific antigen loss [42]. In the last decade, next generation sequencing technology and bioinformatics tools have allowed for important advances in the identification of patient-specific TAAs and the discovery of several neo-antigens that were derived from the high mutational rate of tumor cells [43,44]. The process for neo-antigen identification and for their immunogenicity definition requires time and costs that are often not sustainable in the clinical practice despite the potentiality of personalized DC-based vaccine.

An alternative approach to provide an efficient source of multiple TAAs is the employment of whole tumor lysates [45,46]. In this case, the immunogenicity of the tumor cell can be enhanced ex vivo treating autologous cancer cells with specific chemotherapeutic drugs or physical therapeutic modalities before lysate preparation [47]. Indeed, the mechanism of antigen presentation by DCs and the capacity to activate antitumor immune T cells are both thought to be influenced by how tumor cells are handled. In line with this, the immunogenic potential of UV-irradiation was demonstrated for the first time in 1991 when Begovic et al. showed that UV-irradiated cancer cells that were injected in immunocompetent mice could induce resistance to subsequent re-challenge with live tumor cells, whereas the vaccination effect was lost in immunodeficient nude mice [48]. Subsequently, other groups confirmed that UV-irradiated apoptotic cells could effectively stimulate antitumor immune responses when used for vaccination in syngeneic mouse models [49]. Oxidized and heat-conditioned tumor cells have been explored as TAA sources for vaccine formulation, which resulted in being more immunogenic as compared to tumor lysates obtained through freeze/thaw repeated cycles [50,51,52,53,54]. The ex vivo induction of ICD is a more recent strategy to modify tumor immunological features. This approach provides, at the same time, TAAs and molecules that can act as DC activating signals.

### 2.2. In Situ Vaccination

*In situ* vaccination is an alternative way to exploit the potentiality of DCs for priming tumor-specific T cell activation, that is, the intratumoral inoculation of DC activators/adjuvants, such as TLR agonists [29,55,56] or CD40L [57] to stimulate DCs to uptake and process TAAs and specific neo-antigens directly released from tumor cells in the surrounding TME [58]. Recent preclinical studies and clinical trials combined the use of DC stimulators with the growth factor FMS-like tyrosine kinase-3 ligand (FLT3L) to increase DC numbers in peripheral blood [59,60]. For optimal delivery, the adjuvants can be encapsulated in nanoparticles, liposomes, or immunostimulatory complexes specifically targeting DCs [61,62,63], whereas to guarantee a sufficient availability of immunogenic TAAs, in situ vaccination can be combined with ICD-inducing therapeutic modalities, such as doxorubicine or radiotherapy.

In order to overcome the low number of pre-existing tumor-infiltrating DCs, another possible approach is represented by the intratumoral inoculation of ex vivo generated unloaded DCs, also called in situ DC vaccination. This strategy also benefits from the ability of inoculated DCs to directly uptake multiple TAAs in vivo, obviating the need to generate an ex vivo TAA cargo or to identify and select specific epitopes. Indeed, if antigen identification and their immunogenicity definition are expensive and time-consuming, the preparation of tumor cell lysates is also subject to limitations, among which, primarily, the paucity of autologous tumor cells amenable to ex vivo manipulation. Yu and colleagues showed that only the combination of chemotherapy with in situ DC vaccination induced effective antigen-specific CD8+ and CD4+ T-cell mediated responses in an advanced-stage breast cancer model, whereas neither chemotherapy nor DC inoculation elicited antitumor immune responses when being used as single treatments [64]. Recent clinical trials showed the efficacy of in situ DC vaccination in achieving clinical and immunological responses. In a clinical study, where CCL21 transduced DCs were used in non-small cell lung carcinomas, a significant increase in CD8+ T cell infiltration was detected in 56% of patients and it was associated with PD-L1 up-regulation [65]. Moreover, intratumoral injection of activated DCs in patients with different neoplasms enhanced lymphocyte infiltration and specific cytokine production by DCs, which correlated with stable disease and prolonged survival [66]. Recently, Cox and collaborators investigated the combination of intranodal injection of interferon-conditioned DCs with low-dose rituximab in follicular lymphoma patients. Interestingly, in 50% of patients, objective clinical response was observed not only in primary treated lesion, but also in the untreated ones, highlighting the ability of inoculated DCs to enhance the “abscopal effect” of the treatment [67]. The accumulated experimental evidence strongly supports the idea that in situ DC vaccination benefits from tumor pretreatment with pro-apoptotic agents [64,67,68] and, in particular, with ICD inducers. In fact, in vivo employment of ICD inducers results not only in TAA release by dying cells, but also in the secretion of DC activating DAMPs and more efficient engulfment of tumor cells by DCs [57,58,69,70,71].

## 3. Effects of ICD Hallmarks on Immune Cells in Tumor Microenvironment

The definition of apoptosis as a non-immunogenic, but silent or tolerogenic, physiological process has been increasingly questioned after ICD discovery. In fact, specific anticancer drugs (such as anthracyclines or platinum compounds) and physical therapeutic modalities can promote the modulation of a subset of DAMPs in cancer cells that are capable of inducing both apoptosis and an antigen-specific immune response [72]. Yatim et al. recently introduced the concept of “signal 1” to refer to the activation of cell death pathways as an initiating immunological event, according to the ICD definition [6] (Figure 1). Finally, “Signal 1” relies on the release of constitutive DAMPs (cDAMPs) or the production or modulation of inducible DAMPs (iDAMPs) by dying cells.

The subcellular localization of those DAMPs that associate with ICD is the main determinant of their classification: 1) class 1 DAMPs are exposed on the plasma membrane and include heat-shock protein (HSP) 70, HSP90, and calreticulin (CRT); 2) class 2 DAMPs are secreted extracellularly and include ATP, high-mobility group box (HMGB)-1, and uric acid; and, 3) class 3 DAMPs are generated during end-stage degradation of ICD and they include mitochondrial components, DNA, and RNA [16,18,19].

### 3.1. CRT

CRT is a Ca^2+^-binding protein that is abundant in the lumen of the ER that exhibits several functions, such as the modulation of Ca^2+^ signaling and homeostasis [73]. It is a chaperone of several proteins, in particular the disulfide isomerase Erp57 [74]. In response to ICD inducers, CRT translocation occurs from the ER lumen to the surface of stressed and dying cells. CRT exposure precedes phosphatidylserine externalization and it is followed by Erp57 translocation to the cell membrane [75,76]. This mechanism is triggered by ER-stress-dependent activation of PERK that phosphorylates the eukaryotic translation initiation factor (eiF)-2α, which results in its inhibition. Subsequently, pre-apoptotic cleavage of procaspase-8 occurs with the subsequent activation of the pro-apoptotic proteins BAX and BAK [76].

During ICD, CRT acts as a potent “eat me” signal that facilitates the engulfment of dying cells by specific phagocytic cells (macrophages, neutrophils, and DCs). Cell surface-exposed CRT binds low-density lipoprotein receptor related protein 1 (LRP1, also known as CD91) on macrophage and DC cell membrane, and this interaction is necessary for antigen cross-presentation to CTLs. Moreover, CRT-CD91 interaction triggers the signaling pathway of NF-κB in DCs and the release of a series of pro-inflammatory cytokines in the extracellular milieu leading to Th17 priming [77]. Tumor cells that lost CRT expression undergo apoptosis without being efficiently engulfed by phagocytes, which suggests that CRT exposure is critical for phagocytosis of the dying cells. Moreover, CRT knockdown abolishes the immunogenicity of cancer cells undergoing ICD and inhibits their ability to induce a protective immune response when used in experimental mouse vaccination protocols. On the other hand, the administration of recombinant CRT confers immunogenic properties to cell death processes that are promoted by non-ICD inducers [76,78,79].

In patients with non-small cell lung cancer, CRT expression is correlated with a favorable clinical outcome that is putatively associated to a higher infiltration of mDCs and effector memory T-cell subsets [23]. A recent bioinformatic study revealed a strong correlation between CRT mRNA expression and the local immune infiltrate density and composition. In fact, it demonstrated, in breast, colorectal and ovarian cancers, CRT expression levels that were positively correlated with DC and CTL infiltration ensuring immunosurveillance [80].

The pro-phagocytic function of surface-exposed CRT is antagonized by CD47, a transmembrane protein, also known as integrin-associated protein (IAP), which acts as a “do not eat me” signal. CD47 interacts with signal regulatory protein alpha (SIRPα), inhibiting phagocytosis by macrophages and DCs. The blockade of CD47 with a monoclonal antibody allows for phagocytosis of dying cells [81] also in the absence of CRT exposure [82]. However, the role of CD47 in ICD has yet to be fully elucidated. 

### 3.2. HSP70 and HSP90

Heat shock proteins (HSPs) are a family of molecular chaperones that fold and remodel proteins [83]. During ICD, HSP70 and HSP90 are exposed on the cell membrane of dying cells and they are later released in the TME [84]. Cell surface-exposed HSP70 and HSP90 allow for the interaction of these proteins with their respective receptors, CD40 and CD91, on DCs [15].

HSP70 can stimulate DC maturation by promoting the upregulation of CD86 and CD40 and, through these co-stimulatory signals, augments CD8+ CTL immune responses [85,86]. Moreover, this molecular chaperone interacts with TLR4 on DCs, which activates NF-κB and the subsequent release of pro-inflammatory cytokines [87]. Lin et al. showed that HSP70 is also an important component of shikonin-treated tumor cell lysates (SK-TCL), which are able to promote DC-mediated T-cell proliferation [88].

The interaction of HSP90 with CD91 leads to the activation of DCs and it facilitates the tumor antigen cross-presentation to CTLs; in fact, the blockade of CD91 receptor completely abolishes T cell activation in cross-presentation experimental models [89]. Spisek et al. showed that tumor cells that were treated with bortezomib (a proteasome inhibitor) present increased expression of HSP90, which induces the upregulation of the maturation-associated markers CD80, CD83, and CD86 on DCs [90].

### 3.3. HMGB1

HMGB1 is a nuclear non-histone chromatin-binding protein that regulates the transcription of different proteins, such as NF-κB species and p53, and promotes V(D)J recombination in developing B cells [20,91,92]. In some stress circumstances, HMGB1 moves from the nucleus to the cytoplasm, where it activates autophagy by interacting with beclin 1 [93,94]. Necrotic stressed cells passively release HMGB1 in the extracellular environment in the late stage of ICD, which requires the loss of both nuclear and plasma membrane integrity [95]. An intense inflammatory response is triggered when this DAMP interacts with its specific receptors (TLR2, TLR4, and/or RAGE) on the APC surface. [96,97,98]. The binding of extracellular HMGB1 to TLR4 induces efficient cross-presentation of tumor antigens by DCs. In the absence of HMGB1 or TLR4, dying cells are regularly engulfed by DCs. However, phagosomes fuse with lysosomes, causing the dying cell degradation rather than antigen presentation by DCs [99,100]. Anthracyclines also increase HMGB1 release in EBV immortalized lymphoblastoid B-cell lines and trigger a TLR4-mediated autocrine/paracrine loop resulting in NF-κB activation and the prolonged stimulation of EBV specific T-cell precursors [101]. Finally, HMGB1 activity is regulated by redox modification; indeed, fully reduced HMGB1 acts as a chemoattractant, disulphide-bond possessing HMGB1 is able to produce pro-inflammatory cytokine, and fully oxidized HMGB1 results in being inactivated [102,103].

### 3.4. ATP

During ICD, the dying cells release ATP in the extracellular milieu and it acts as a “find me” signal attracting DCs, monocyte, and neutrophils. Extracellular ATP interacts with ionotropic (P2X7) and metabotropic (P2Y2) purinergic receptors on APC, promoting chemotaxis and differentiation, respectively [24,104]. Several pro-inflammatory processes, such as the activation of the NLR family, pyrin domain containing 3 (NLRP3) inflammasome, and secretion of IL-1β, are promoted when ATP activates DCs through the P2X7 receptor. IL-1β secretion and antigen presentation are required for promoting the polarization of interferon-γ (IFNγ)-producing CD8+ T cells. Ghiringhelli et al. have shown that mice lacking NLRP3 or P2RX7 seem to be unable to develop efficient adaptive immune responses during ICD [24,105].

ATP release occurs through different mechanisms, depending on the apoptotic stage and the type of stimulus. Pre-apoptotic release of ATP depends on the PERK-regulated, proximal secretory pathway and phosphatidylinositol 3-kinase (PI3K)-dependent exocytosis. In the early apoptotic stage, ATP can be released through either pannexin 1 (PANX1) or in an autophagy-dependent manner, depending on the cell death stimulus. Conversely, in the late stages of apoptosis, ATP is passively externalized due to membrane permeabilization [78,106,107].

In the context of ICD, ATP is secreted through sequential events that are triggered by caspase activation, which include: 1) ATP accumulation inside autolysosomes that is associated to autophagy; 2) translocation of LAMP-1 to plasma membrane; 3) cellular blebbing dependent on ROCK1; and, 4) the opening of PANX1 channels [106,108,109].

On the other hand, extracellular ATP can be reduced by the action of surface-expressed ectonucleotidases, such as CD39 and CD73 Treg and Th17 cells, which promote immunosuppressive events and tumor progression, express CD39.

Altogether, these data indicate that ATP is indispensable for tumor cell immunogenicity [110,111].

## 4. Tumor ICD Handling in DC-Based Vaccine Development 

The induction of ICD is justified by the idea that immunogenic apoptotic cells are a proper in vitro source of both antigens and adjuvants for efficient ex vivo DC activation and tumor-specific T-cell stimulation [69,91,112]. Along this line, we have recently performed a review of the literature regarding the modulation of ICD-executed DAMPs and its impact on immature and mature DCs [113]. We conclude that linking the immunogenic potential of ICD with DC-based vaccination is a promising approach that could achieve future translational success. Here, we examine the data from recent studies examining the effectiveness of ICD inducing therapies in the preparation of DC vaccines in suitable preclinical and/or clinical models (Table A1).

### 4.1. Chemotherapy

The combination of chemotherapy and immunotherapy has been generally discouraged because of the immune suppressive side effects of most cytotoxic agents. However, it has now become evident that the immune effects of several chemotherapeutics, including anthracyclines, platinum derivatives, alkylating agents, and proteasome inhibitors, may be beneficial for the patient [69]. Among them, some well-known ICD inducers, such as doxorubicin, interferon-α, retinoic acid, colchicine, and shikonin, have been widely tested in DC-based vaccination protocols [113]. In the following sections, we recapitulate the data from studies employing chemo-ICD-inducers to improve the immunogenicity of tumor cells, and their suitability as an antigen and adjuvant cargo in prophylactic and therapeutic DC-based immunization regimens.

#### 4.1.1. Doxorubicin

Doxorubicin (Dox) is an antibiotic that is derived from *Streptomyces peucetius*; topoisomerase inhibition and DNA-intercalation are its best-known cytotoxic effects [114]. This anthracycline has been widely used as a chemotherapeutic agent and several studies have shown its intrinsic ability to trigger ICD and the spatiotemporal release of DAMPs from dying cells [115,116]. The administration of apoptotic tumor cell lysates following Dox treatment as antigenic cargo for DC-based vaccine was recently explored in both prophylactic and therapeutic settings [117].

Bone marrow (BM)-derived DCs were loaded with Dox-treated, or its corresponding control, and then intradermally injected into syngeneic mice. Mice receiving Dox-treated cell lysate-loaded DCs (prophylactic assay) exhibited significantly reduced tumor development ability. However, this protective effect was not observed in a therapeutic vaccination regimen. This was reversed by combining the immunotherapy approach with the virally delivered CXCR4 antagonist, which generated a more permissive environment for the induction of antitumor immunity [117]. These preliminary data suggest that the efficacy of the vaccine is highly dependent on the pre-existing immunomodulatory network in the TME.

#### 4.1.2. IFN-α

Immunotherapy that is based on the exogenous administration of IFN-α is approved as an adjuvant treatment for melanoma. High-dose IFN-α treatment has long been known to prolong patient survival following resection of the primary lesion [118]. Unfortunately, this regimen is associated with serious adverse effects, such as fatigue, myalgia, pyrexia, and depression. Clinically, the majority of patients with objective clinical response also showed functional suppression of Treg lymphocytes, decreased migration of DCs to the tumor site, and increased activity of effector CD8+ CTLs [119]. While lower doses may decrease these non-desirable consequences, they do not offer the same therapeutic outcome [120].

We have recently proposed the use of a 9-cis-retinoic acid (RA)/IFN-α combination as a novel and highly effective modality to induce ICD ex vivo in aggressive and/or refractory lymphoma cells for the optimization of DC-based vaccines [121,122]. The RA/IFN-α combination stimulated autophagy [123] and induced high levels of apoptosis in vitro [124], and highly immunogenic tumor cell lysates that were generated following this treatment were successfully used for DC loading (100 μg of tumor lysate for 10^6^ DCs). To this end, mo-DCs were ex vivo differentiated in the presence of GM-CSF and IFN-α. RA/IFN-α-treated tumor cells boosted mo-DCs maturation and activation. Therapeutic vaccination of tumor-bearing mice with mo-DCs loaded with RA/IFN-α-tumor cell lysates inhibited tumor growth, which was associated with a systemic Th1-skewed response. Our results were in agreement with the discovery that the activation of DCs mediated by Dox- [116] or photodynamic therapy (PDT)-killed tumor cells [125] is partially mediated by autocrine and/or paracrine pathways that are triggered by IFN-α. Along this line, a variety of stimuli that promote endogenous type I-IFN expression are currently evaluated as an adjuvant strategy in vaccine protocols that aimed at triggering and enhancing immune system activation. In fact, the induction of IFN-α/β increases cellular immunity and, therefore, represents a promising solution to improve the immunogenicity of cancer vaccines, differently from traditional adjuvants, such as aluminum compounds, which predominantly encourage humoral immune responses [126,127].

#### 4.1.3. Shikonin

Shikonin (SK) is a phytochemical that is isolated from the root tissues of the traditional medicinal herb *Lithospermum erythrorhizon*. SK exhibits several antitumor pharmacological properties [128]. In addition, SK promotes ICD that is associated to autophagy-dependent DAMP mobilization [88,129,130]. Initial studies have compared the effectiveness of SK and Dox, among other chemotherapeutic, in protocols making use of immunogenic tumor cell lysates. They demonstrated that, following TLR4-dependent stimulation with LPS, lysates from both Dox and SK-treated tumor cells enhanced the maturation status of BM-derived DCs, concomitant with the priming of Th1/Th17 effector cells. In a therapeutic protocol, both of the vaccines retarded tumor growth, prolonged survival rate [129], and suppressed metastatic spread [88,130]. Furthermore, this DC-based vaccination partially blocked the level and progression of the rapamycin-promoted metastatic side effect, which was associated to Treg cells and myeloid-derived suppressor cells (MDSCs) activation [131].

#### 4.1.4. Colchicine

Colchicine exhibits antitumor properties via the inhibition of tubulin polymerization [132], and it also has the ability to induce the expression of ICD-related molecules in targeted tumor cells. Immunogenic lysates from colchicine-treated tumor cells were tested as the cargo of BM-derived DCs. The study introduced an interesting novel methodology for DC differentiation, consisting of the addition of the cytokines GM-CSF and IL-4 to the culture medium during the entire co-culture incubation period. This modification significantly improved the efficacy of the proposed vaccine. This regimen enhanced therapeutic immunity in a tumor-challenged animal model [133].

### 4.2. Physical Therapeutic Modalities

Although ICD was initially conceived as a chemotherapeutic-based tumor cell death, physical anticancer modalities have shown comparable immunogenic potential that can be exploited in DC-based vaccine protocols. These antitumor modalities include high hydrostatic pressure (HHP), PDT, radiotherapy (RT), ultraviolet light (UV-light), and hyperthermia (HT). In fact, these physical therapeutic modalities might be desired over chemotherapeutically induced ICD for preparing DC-based vaccines, since the former do not leave residues of active drugs [13]. Here, we summarized the current state of knowledge regarding the preclinical and clinical use of tumor cells that were killed by physical ICD inducers, in the formulation of DC-based anticancer vaccines.

#### 4.2.1. High Hydrostatic Pressure

HHP is a non-thermal process that has shown the ability to induce cell death in murine and human cells in a range of 100–400 MPa, which triggers cell rounding, cytoplasmic gelation, enzymatic inhibition, and the suppression of protein synthesis, but without disturbing DNA integrity [13,134,135,136].

Preliminary results have demonstrated that apoptosis induction by HHP was accompanied by exposure/release of several immunogenic molecules, such as HSP90, HSP70, CRT, ATP, and HMGB1, in human prostate, leukemia, and ovarian tumor cell lines and/or primary tumor cells [137,138]. For that reason, the use of HHP-killed human tumor cells has been tested for DC-based vaccine generation [137,138,139,140]. In these assays, human mo-DCs were co-cultured at a DC/tumor cell ratio of 5:1 with cancer cells that were previously subjected to lethal doses of HHP, and then incubated with poly(I:C) or LPS as maturation stimuli. It was demonstrated that HHP increased the rate of phagocytosis of tumor cells by DCs. Moreover, DCs that were loaded with HHP-treated tumor cells exhibited an upregulation of maturation markers, chemotactic migration, and the release of pro-inflammatory interleukins when compared to unloaded DCs. In addition, HHP-treated tumor lysate loaded-DCs were able to activate tumor-specific T cells while also downregulating the number of Treg [137,138,139,140].

These achievements spurred the use of DCs loaded with HHP-treated tumor cells in in vivo preclinical studies. Prophylactic and therapeutic immunization regimens were performed. This approach involved the ex vivo differentiation of DCs from murine BM precursors, their co-culture with 200 MPa-HHP-treated tumor cells, and, finally, their incubation with CpG ODN 1826 (TLR9 agonist) as a maturation stimulus. Immunogenic and weakly immunogenic cell lines were used as tumor models. In prophylactic protocols, the mice were immunized twice with DC-based vaccines and then challenged with tumor cells. Both HHP-treated tumor models promoted a higher cytotoxic effect and IFN-γ production from spleen effector cells when used as cargo of DCs when compared to immunization with tumor cells alone. However, only HHP-treated immunogenic tumor cell-pulsed matured DCs significantly slowed down the growth of transplanted syngeneic tumors. The therapeutic efficacy of these DC-based vaccines was evaluated in combination with chemotherapy with docetaxel. In this regimen, DC-based vaccines were administered after the tumor challenge at regular intervals between docetaxel administrations. The tumor inhibitory effect of both immunogenic and weakly immunogenic models resulted in more effective suppression of growth, when compared with the untreated tumor-bearing animals [138]. A similar outcome was reported when the DC-based vaccines pulsed with HHP-treated tumor cells was combined with the chemotherapeutic cyclophosphamide [140]. In addition, the ability of DC-based vaccines to inhibit tumor recurrence was demonstrated in clinically relevant models of minimal residual tumor disease after surgery of both immunogenic and weakly-immunogenic models [140].

Collectively, these data introduce HHP as a potent ICD inducer, which can generate an effective antigen/adjuvant tumor source for DC-based vaccine, for its application in prophylactic, combination-mediated therapeutic, and adjuvant anticancer therapy.

#### 4.2.2. Photodynamic Therapy

PDT, a physical modality that is approved by regulatory agencies for the treatment of cancerous and other lesions, is a photochemistry-based two-stage procedure. Photochemical processes are generated upon irradiation of tumor sites with a PS-exciting light of specific wavelength after administration of the photosensitizer agent (PS). PS toxicity is not appreciated unless oxidative stress occurs in response to its activation [125,141]. In addition to eliciting direct cytotoxic effects [142,143,144,145,146,147] and vascular damage [148], PDT also induces immunological reactions [20,149,150,151]. PDT outcome has been associated with the subcellular localization of PSs [146,152,153,154] and the PDT dose delivered, leading to tumor cell photodamage through four, not mutually exclusive, cell death pathways, such as apoptosis, necrosis, autophagy, and paraptosis, which could differentially stimulate the host immune system [155]. DAMP exposure that was crucial for ICD was reported following photoactivation using several photosensitizers [125,156,157,158]. The main PDT-generated DAMPs that are involved in ICD include CRT, ATP, HSP90, HSP70, and HMGB1 [84,91,109,125,149,150]. In a study on skin squamous cell carcinoma by Ji et al, immunogenic apoptotic tumor cells induced by Aminolevulinic acid (ALA) mediated PDT could enhance the antitumor activity of DC-based vaccines in mice. The functional DC maturation and ability to promote T cell proliferation relied on the induction of higher levels of IFN-γ and IL-12 by apoptotic tumor cells that were attained by ALA-PDT relative to necrotic cells that result from PDT or freeze/thaw cycles. Thus, ALA-PDT-DC vaccines provided effective adaptive immune system responses while reducing the production of the immunosuppressive cytokine IL-10 [159]. A preclinical study using hypericin (Hyp-PDT) as single-agent ICD inducer for DC-based vaccines in an orthotopic high-grade glioma (HGG) mouse model showed a distinctive survival benefit in the prophylactic or curative setups. The immunogenicity of the vaccine was determined by the generation of ROS and danger signals, such as extracellular HMGB1, extracellular ATP, and surface CRT. DC vaccines produced a Th1-biased immunity, which was associated with more favorable patient prognosis [160]. Consistently, Hyp-PDT triggered tumor-specific immune responses in Lewis lung carcinoma (LLC). In this study, PDT-treated LLCs (PDT-LLC) and DCs co-cultured with PDT-LLCs (PDT-DC) both elicited potent antitumor responses in mouse models. The membrane dysregulation of DAMPs and CD47 underlies these tumor-specific immune responses and the ensuing clinical efficacy [161]. A versatile nanoformulation was developed in order to optimize the availability of TAAs and DC recruitment. Chimeric cross-linked polymersomes (CCPS) with co-encapsulated Dox and 2-(1-hexyloxyethyl)-2devinyl pyropheophorbide-a (HPPH) as photosensitizer (CCPS/HPPH/Dox) were tested to develop an in situ DC vaccination using copolymers as adjuvants combined with TAAs for MC38 colorectal cancer treatment. A single low-dose administration of this Dox and HPPT combination could inhibit primary and distant MC38 tumor growth [71]. PDT has been associated with expression of several DAMPs that are involved in ICD, as reported above. Recently, we explored the association of PDT with the type I IFN pathway, which can enhance cell mediated immunity [126]. We demonstrated that the incubation of B16-OVA melanoma cells with Me-ALA as precursor of the endogenous photosensitizer, protoporphyrin IX (PpIX), induced oxidative ER-stress mediated-apoptotic cell death and upregulation of type I IFN expression upon exposure to visible light. DCs pulsed with PDT-treated tumor cells exhibited an enhanced expression of co-stimulatory signals (CD80, MHC-II) and tumor-directed chemotaxis [125]. Overall, accumulating preliminary evidence highlights the future potential relevance of PDT in adoptive immunotherapy protocols.

#### 4.2.3. Radiotherapy

Radiotherapy is a cancer treatment modality that relies on the anti-neoplastic activity of irradiation (X- or γ-rays), which induces DNA damage and apoptosis of tumor cells [13]. Two preliminary prophylactic preclinical studies demonstrated that irradiated tumor-primed DCs were highly effective in the prevention of local tumor outgrowth [162,163], even before the concept of ICD was introduced [115]. This immune protection was dependent on proper CD4+ and CD8+ T cell activity [162]. Presently, RT is defined as an ICD-inducer that is associated with oxidative and collateral ER-stress [20]. A recent study demonstrated that X-ray treatment enriched protein carbonylation in tumor cell lysates. For vaccine generation, DCs were incubated with tumor cell lysates (2 mg protein per 10^6^ DCs), and then cultured in GM-CSF and LPS containing medium. In a prophylactic setting, this preparation induced a reduced infiltration of Tregs, tumor-associated macrophages (TAMs), and MDSCs concomitant with enriched populations of CD3+ T cells and a stronger infiltration of Th1 cells and CTLs. The carbonylation degree positively correlated with the survival rate of vaccinated tumor-challenged mice, even when DC activation was not triggered [47]. Taken together, these data strongly suggest that X-ray irradiated lysates may work as an effective antigen/adjuvant source in clinical DC vaccination studies.

#### 4.2.4. UV-Light

Ultraviolet light (UV) is an electromagnetic radiation that can be classified, from lowest energy/longest wavelength, in UVA, UVB, and UVC. In cells, UV light disturbs DNA by generating two major types of photoproducts: cyclobutane pyrimidine dimers (CPDs) and pyrimidine-pyrimidone (6-4) photoproducts [(6-4) PPs]. The consequence of this DNA damage is the permanent arrest of DNA transcription and/or replication, which finally leads to cell death [164]. Interestingly, UV-light has been associated to the induction of ICD [13], which was extensively exploited in the generation of antigenic cargo for vaccine development.

Initial reports in murine models documented the prophylactic [165] and therapeutic [166] effects of UVB-mediated apoptotic tumor cell-pulsed DCs. These studies used a modified method for the isolation of DC-precursors from BM, which generated a fully matured DCs pool before co-culture with tumor cells that were previously irradiated with UVB-light (at an approximate tumor to DC ratio of 1:2). For that reason, it was not possible to discriminate the degree of maturation of DCs that were induced by the isolation method and/or the dying tumor cell stimulus with certainty [165,166]. However, it was demonstrated that UVB treated tumor cell primed-DCs exhibited an increased secretion of IL-12 [165]. These activated DCs promoted significant tumor extinction or no tumor formation in the prophylactic assay [165], but they failed to retard tumor growth in the therapeutic regimen, which was partially recovered in combination with IL-2 exogenous administration [166]. In a clinical setting, UVB was also used to generate apoptotic tumor cells from enzymatically dissociated surgical specimens of neck squamous cell carcinoma primary lesions or metastatic lymph nodes. The DCs were differentiated from autologous monocytes, which were obtained from leukapheresis, and then co-cultured ex vivo with UVB-treated whole tumor cells, used as a source of tumor-associated epitopes. This DC-based vaccine was well tolerated without significant toxicities. Unfortunately, the pilot study failed to enroll the proposed number of patients. Notwithstanding this limitation, the trial showed that vaccinated patients were mounted an effective antitumor immune response and exhibited long term disease-free survival [167].

Increased knowledge of the phenotype and function of DC subsets allowed for the development of next-generation DC vaccines, thus making use of primary circulating DCs from cancer patients. In particular, human cDC1s and their mouse equivalents constitute the major DC subset for the initiation of anti-tumor immunity. In fact, this population exhibits superior ability in the uptake of dying or dead cell material, processing of cancer cell-associated antigens for cross-presentation, and the secretion of IL-12, a crucial cytokine for the activation of anti-cancer CD8+ cytotoxic T-cells [168,169,170]. A recent study demonstrated the feasibility and efficacy of the vaccination with mouse cDC1 cells pulsed ex vivo with an autologous UVC-killed tumor cell lysate. The mouse DCs were obtained from the spleens of mice that were grafted with a genetically modified clone of B16 melanoma that secretes FMS-like tyrosine kinase 3 ligand (FLT3L), leading to the enrichment of the circulating cDC1 pool. Primary cDC1 were co-cultured with UVC-treated tumor cells in order to generate the mature cDC1 vaccine. These tumor cell preparations secreted significantly more HMGB1 than the untreated counterparts, and their lysates promoted the expression of maturation markers in cDC1s. Such DC-based vaccine was intravenously or intradermally injected into mice. This treatment enhanced the presence of tumor-reactive CD8+ and CD4+ T cells in primary tumors and tumor-draining lymph nodes. Moreover, in a prophylactic approach, vaccination with UVB-killed tumor cell-loaded cDC1s protected against melanoma engraftment. The therapeutic immunization limited the tumor growth in three different mouse cancer models, two of which do not express exogenous or dominant antigens. Tumor cell lysate-loaded cDC1 administration significantly increased the efficacy of anti-PD-1 therapy in both immune checkpoint antibody sensitive and resistant models [171].

Altogether, these data demonstrated the impact of UV-light as an ICD inducer on tumor cells and the ability of this cargo to enhance dendritic cell activation and improve the outcome of clinical and pre-clinical DC-based prophylactic and therapeutic regimens.

#### 4.2.5. Hyperthermia

HT involves the exposure to temperatures that range from 41 °C to 44 °C [172]. It has been reported that lower temperatures (41–43 °C) cause apoptotic cell death in tumor cells, whereas necrosis is observed at higher temperatures (>43 °C) [173]. Cell damage was shown to be dependent on thermal dose, which is a combination of the time and temperature of exposure, and on cell type, being characterized by changes of cytoskeletal organization, the disruption of fluidity and stability of cellular membranes, inhibition of transmembrane transport proteins and cell surface receptors, reduced synthesis of RNA and DNA, DNA-damage, and aggregation of denatured proteins [174]. Moreover, HT has the capacity to induce cytotoxicity of cancer cells and prime both innate and adaptive immunity. For that reason, this treatment was used as stress-promoting strategy in combination with other chemotherapeutic or physical antitumor agents with the aim of generating immunogenic whole tumor cell lysates [175,176,177,178].

An interesting approach was initially designed to prove the DC-stimulating ability of heat stressed tumor cells subjected to apoptosis. Mouse leukemia cells 12B1-D1 were genetically modified to undergo Fas/Fas ligand-dependent apoptosis in response to the chemotherapeutic AP20187. Thermo-stressed tumor cells were co-cultured at a 1:1 ratio with syngeneic BM-derived DCs and then incubated with AP20187 (nontoxic to DCs). These DCs exhibited a mature phenotype, which was characterized by an up-regulation of co-stimulatory molecules on their surface, the production of the pro-inflammatory cytokine IL-12, and the substantial enhancement of immunostimulatory function in a mixed leukocyte reaction [176]. Moreover, they promoted the rejection of co-injected viable leukemia cells, and also established long lasting specific immunity and protection against re-challenge [177]. Interestingly, silent apoptotic death only induced by AP20187 failed to alert the immune system. In fact, the study demonstrated that HT-stressing the tumor cells before triggering apoptosis was required for the membrane exposure of HSP72 and HSP60 [176,177], hence supplying danger signals that may be critical for the generation of tumor-specific immunity. Conversely, BM-DCs did not express higher levels of activation markers when murine melanoma cells were lethally stressed by heat shock and γ-irradiation and then used as immunogenic cargo. Despite this, they induced a significant retardation of tumor growth when applied as a prophylactic vaccine. Treg depletion enhanced the success of this prophylactic protocol, allowing for the improvement of long-lasting tumor protective immunity [178].

In line with these results, the sequential combination of hyperthermia, γ-irradiation, and UVC was selected as the method of choice for preparing whole tumor lysate-loaded DC vaccines in a therapeutic clinical regimen against indolent non-Hodgkin lymphomas [175]. Eighteen patients who had relapsed after at least one chemoradiotherapy regimen were enrolled in this pilot study. The tumor cells were isolated from lymph nodes and/or peripheral blood, whereas the DCs were differentiated from autologous monocytes. A mature mo-DCs phenotype was observed in more than 90% of the APCs after loading the immature mo-DCs with killed and heat-shocked tumor cells. The patients were subjected to four doses of subcutaneous host-specific vaccination, which was well tolerated without autoimmune reactions. The adverse effects were minimal, consisting of injection site reactions of grade 2 or lower (mild to moderate). Interestingly, 33.3% of the patients achieved significant objective clinical responses—16.7% lasting complete responses and 16.7% partial responses—and 44% exhibited stable disease. The success of this therapeutic regimen was associated with significant immune modulation, including a reduction in Treg, an increase in natural killer cells, maturation of lymphocytes to the effector memory stage, and production of IFN-γ and IL-4 by circulating T cells in response to tumor-specific antigen encoded peptides [175]. Remarkably, a positive correlation was demonstrated between CRT and HSP90 surface expression in the DC antigenic cargo and the effectiveness of the achieved clinical and immunologic responses [179].

Overall, these reports demonstrated that hyperthermia effectively enhances the immunogenic potential of tumor cells mainly by the mobilization of DAMPs prior to cell death-associated events. The combination of thermo-stress with apoptotic/necrotic stimuli confers efficient antitumor capabilities to DCs, which could be applied in prophylactic and therapeutic immunization protocols.

## 5. Conclusions

From an extensive review of the literature, it appears that multiple aspects should be considered when designing DC-based vaccines. As shown in the present review, some of these, among which DC origin, differentiation and culture method, maturation stimuli, antigen selection and loading, and route of administration, are variably combined in the diverse protocols analyzed (Table A1). The lack of a univocal protocol hampers the definition of standard procedures, but, above all, does not allow for a realistic comparison among different preclinical and clinical studies. Regardless of the regimen applied, DC vaccination is characterized by a very favorable toxicity profile [180]. Nevertheless, several clinical studies have pointed to the limitations of DC-based protocols despite the ability of DC-based vaccines to elicit measurable immunological responses, whereby this approach fails to generate effective and lasting clinical antitumor responses, especially when used as monotherapy [181]. Among the main obstacles, the immunosuppression that is associated to TME related-factors appears to be extremely relevant. Therefore, the combination of DC vaccination with appropriate anticancer treatments that are able to modulate the TME toward a pro-inflammatory/stimulatory status could improve its efficacy. In line with these considerations, herein we discussed the opportunity to exploit ICD therapeutic inducers to optimize DC-based vaccination. Several preclinical studies employing ICD-DC-based vaccines indicated that an ICD-derived cargo could indeed boost the stimulation of tumor-specific CTLs to achieve efficient tumor control. Particular attention has concentrated in the last years on the opportunity to exploit whole tumor lysates obtained from cancer cells that were pretreated in vitro with ICD inducers as both a personalized source of endogenous tumor antigen and an adjuvant source for proper DC cargo in therapeutic vaccination protocols (Figure 2A). However, the activation of anticancer immune responses following tumor treatment with ICD inducers, clearly demonstrated in several in vivo experimental models, has not been as apparent in the clinical studies performed so far. In addition, in several instances, the promising results that were obtained in prophylactic settings in animal models have not been confirmed once transposed to the therapeutic setting. One possible explanation is that mo-DCs that were obtained from cancer patients can express an altered phenotype, being characterized by a lower expression of co-stimulatory molecules and reduced ability to present antigens to T cells [182]. Furthermore, their maturational capacity can be affected, whereby mo-DCs may induce tolerogenic responses, regardless of the DC maturation stimulus [183,184,185]. On the other hand, it has been documented that patient-derived mo-DC functional bias is transient and tumor-dependent; in a few instances, ex vivo generated patient mo-DCs were indeed shown to recover a level of stimulatory activity that was comparable with healthy donor DCs after cancer surgery [186]. These observations suggest that the timing for apheresis could be crucial for ex vivo development of mo-DCs.

DC-based vaccination could perhaps be more promising as an adjuvant alternative. While surgical resection with curative intent is aimed at removing the entire tumor burden, in an adjuvant setting, where residual occult disease possibly represents a low tumor burden, a DC-based vaccine could be a valid tool to control recurrence and/or metastasis development. 

Additionally, the possibility to integrate in vivo cancer treatment with ICD inducing therapies with in situ DC-vaccination (Figure 2B) or in vivo targeting DC awaits further investigation [71]. In this setting, the inoculation of ex vivo generated autologous mo-DCs could boost the immune-related properties of ICD inducers and improve TAA presentation and CTL stimulation to switch the TME from immunosuppressive to immunostimulatory.

These observations indicate that some limitations need to be considered and overcome in the near future to effectively translate ICD-DC-based vaccines into the clinical setting. In situ or ex vivo ICD appears to be a promising tool for generating tumor cell lysates for DC cargo. A thorough characterization of the molecular and biological events that are involved in this phenomenon should be performed to identify the areas for improvement and ease the translation of this approach to the clinical practice. At the same time, this in-depth exploration could help to identify predictive biomarkers for DC vaccine response, which might potentially assist in the selection of the most proper treatment in different settings.

## Figures and Tables

**Figure 1 pharmaceutics-12-00256-f001:**
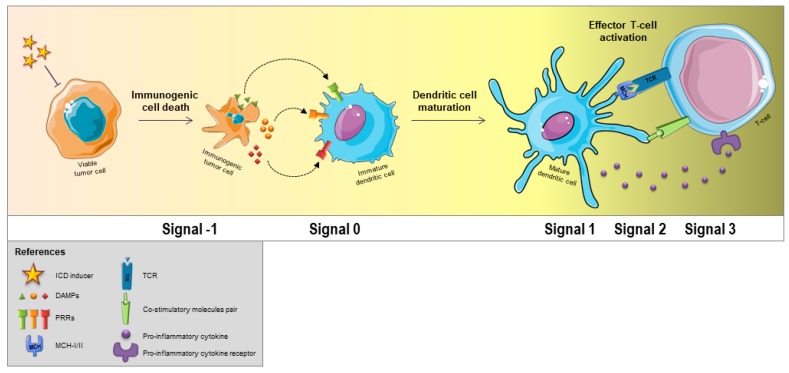
Sequential events required for a proper T-cell mediated antitumor immune response initiated by dying cells. The recently proposed “signal −1” consists of the induction of immunogenic cell death (ICD) on tumor cells. ICD causes the release or surface exposure of a series of damage-associated molecular patterns (DAMPs) through a well-defined spatiotemporally scheme, which activate pattern recognition receptors (PRRs) in immature dendritic cells (DCs), a process that is referred to as “signal 0”. Next, “signal 1” involves the antigen-recognition event that is mediated by the binding of T cell receptor (TCR) to MHC-peptide complexes on the DCs surface. Co-stimulatory signaling is the so-called “signal 2”, whereas “signal 3” is the polarizing and differentiation signal delivered by DCs to T cells, which directly influences their differentiation into antitumor effector cells. Figure created using Servier Medical Art images (http://smart.servier.com).

**Figure 2 pharmaceutics-12-00256-f002:**
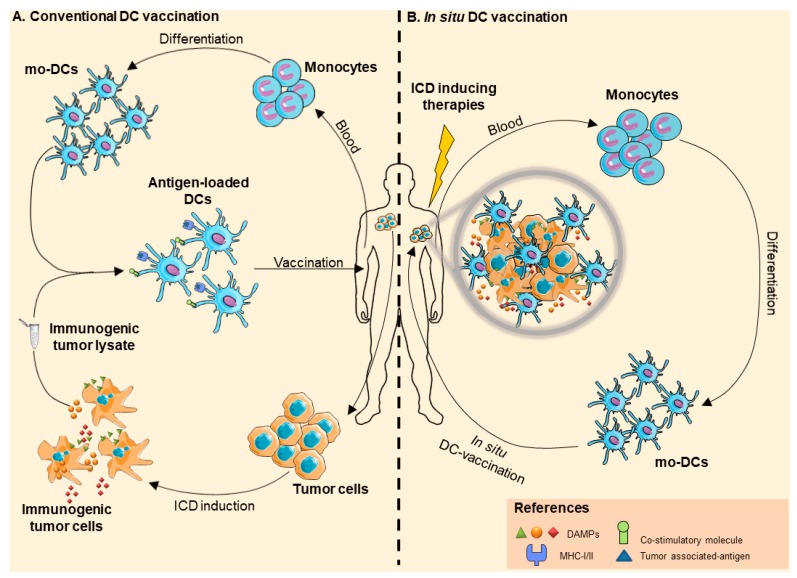
**Different strategies that combine ICD induction and DC-based vaccination**. (**A**) Exploiting conventional DC vaccination in an ICD context consists of the development of ex vivo tumor cell lysate-loaded autologous DCs that are administered systemically in order to trigger tumor-specific immune responses. In this case, ICD is induced ex vivo in patient cancer cells from which tumor lysates are prepared. (**B**) In situ DC vaccination is based on the local administration of ex vivo generated autologuos mo-DCs into the tumor site. In this context, the patient is pre-treated with an ICD-inducing therapy in order to achieve in vivo immunogenic apoptosis and consequent expression of tumor-associated antigens (TAAs) and DAMPs. Intratumor injection of mo-DCs allows in situ TAA presentation and tumor-specific cytotoxic T lymphocyte (CTL) activation, resulting in an expanded antigenic repertoire of T lymphocyte responses. Some images in the figure were created using Servier Medical Art images (http://smart.servier.com).

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
