# Peer review of "Dendritic Cells and Immunogenic Cancer Cell Death: A Combination for Improving Antitumor Immunity"

_pharmaceutics, 2020, doi:10.3390/pharmaceutics12030256_

Round 1
Reviewer 1 Report
This is a very well written summary of current therapeutic approaches using a combination of dendritic cells and immunogenic cancer cell death, with extensive literature resources and comprehensive supplementary images and tables. I believe it will be a useful review for researchers working in this area to treat cancer using this approach. Therefore, I support the publication of the review in Pharmaceutics.
Author Response
Reviewer 1 comments:
This is a very well written summary of current therapeutic approaches using a combination of dendritic cells and immunogenic cancer cell death, with extensive literature resources and comprehensive supplementary images and tables. I believe it will be a useful review for researchers working in this area to treat cancer using this approach. Therefore, I support the publication of the review in Pharmaceutics.
We thank the Reviewer for appreciating our manuscript.
Following the suggestions of all three reviewers we modified some parts of the original review, especially the section 2 (DC-based immunotherapies) and section 5 (conclusion).
Please find enclosed the revised manuscript
Sincerely,
Jessica Dal Col
Reviewer 2 Report
This manuscript reviewed the immunogenic cell death (ICD) in combination with DC-based immunotherapy against cancer. The authors firstly introduced DC-based anti-cancer vaccines and the source of tumor antigen, then summarized typical ICD hallmarks and their effect on immune cells, and lastly discussed ICD-inducing therapies classified by chemotherapy and physical therapy. It was also stressed that the combined treatment of ICD induced in vivo and in situ DC vaccination might be an opportunity in cancer management. The manuscript was written well, and the main idea was to the point, making it a useful reference for researchers in this field. However, some issues should be addressed by authors.
Section 2 only contained one subtitle of "2.1 DC-based vaccines", so these two titles can be combined into one. The manuscript mainly focused on the introduction of ICD hallmark (ICD, HSP, HMGB1, ATP) and ICD-inducing therapies that are linked to DC vaccine preparation. Even though the studies on combining ICD and DC-based vaccination were thoroughly listed, it lacked a comparison among these therapeutic methods. Most of all, the "in situ DC vaccination", which was highlighted in the Introduction part, was rarely mentioned in the following part. In 2.1, it was mentioned that “in situ DC vaccination” could specifically overcome the two obstacles regarding cancer immunotherapy. It would be better to provide examples to support this argument since the importance of in situ DC vaccination was repeatedly stressed in the following content. In addition, how these two obstacles managed to influence the DC-based vaccination at present also requires detailed depiction. In 4.1, several other chemotherapeutics are also reported to induce ICD, such as oxaliplatin and paclitaxel. Is there a specific reason why these four drugs are selected for discussion? It would be better to add relevant information about other frequently-used chemotherapeutics in ICD induction. It is suggested to add the following information in Conclusion: 1) The problems of current DC-based vaccination; 2) the main obstacles that hinder the clinic transformation of DC-based vaccination; and 3) the development potential of DC-based vaccination. The recently published review and research articles should be cited and discussed in the revision, for example, Advanced Functional Materials 2019, 29 (49).Author Response
Reviewer 2 comments:
This manuscript reviewed the immunogenic cell death (ICD) in combination with DC-based immunotherapy against cancer. The authors firstly introduced DC-based anti-cancer vaccines and the source of tumor antigen, then summarized typical ICD hallmarks and their effect on immune cells, and lastly discussed ICD-inducing therapies classified by chemotherapy and physical therapy. It was also stressed that the combined treatment of ICD induced in vivo and in situ DC vaccination might be an opportunity in cancer management. The manuscript was written well, and the main idea was to the point, making it a useful reference for researchers in this field. However, some issues should be addressed by authors.
Please find enclosed the revised manuscript.
Point 1. Section 2 only contained one subtitle of "2.1 DC-based vaccines", so these two titles can be combined into one. The manuscript mainly focused on the introduction of ICD hallmark (ICD, HSP, HMGB1, ATP) and ICD-inducing therapies that are linked to DC vaccine preparation. Even though the studies on combining ICD and DC-based vaccination were thoroughly listed, it lacked a comparison among these therapeutic methods. Most of all, the "in situ DC vaccination", which was highlighted in the Introduction part, was rarely mentioned in the following part. In 2.1, it was mentioned that “in situ DC vaccination” could specifically overcome the two obstacles regarding cancer immunotherapy. It would be better to provide examples to support this argument since the importance of in situ DC vaccination was repeatedly stressed in the following content. In addition, how these two obstacles managed to influence the DC-based vaccination at present also requires detailed depiction.
We appreciated the relevant point stressed by the Reviewer regarding section 2 and, in particular, in situ DC vaccination. We believe that to strengthen the immunogenic effects of ICD inducing therapies with DC injection (or in vivo stimulation) could be a valid therapeutic option to completely eradicate the tumor and prevent recurrences. Thus, in response to the Reviewer's comment we expanded our discussion of "in situ vaccination" in a new section 2.2
Point 2. In 4.1, several other chemotherapeutics are also reported to induce ICD, such as oxaliplatin and paclitaxel. Is there a specific reason why these four drugs are selected for discussion? It would be better to add relevant information about other frequently-used chemotherapeutics in ICD induction.
We thank the Reviewer for this important comment. As the Reviewer pointed out, several chemotherapeutics, including anthracyclines, platinum derivatives, alkylating agents, and proteasome inhibitors, have been reported to induce ICD. Among them, doxorubicin, interferon-α, colchicine, and shikonin have been used to generate tumor cell lysates for dendritic cell-based vaccines. For that reason, in this review, we have summarized only the results from studies of these specific ICD inducers. This is now explicitly stated in section 4.1.
Point 3. It is suggested to add the following information in Conclusion: 1) The problems of current DC-based vaccination; 2) the main obstacles that hinder the clinic transformation of DC-based vaccination; and 3) the development potential of DC-based vaccination.
We appreciate the Reviewer’s encouragement and the opportunity to revise and improve our manuscript. Following the Reviewer´s comments, we have expanded the discussion section highlighting some of the issues associated to current DC-based vaccination protocols, some of the limitations that need to be overcome for translating these protocols from the preclinical to clinical setting, and discussing how ICD could improve the development of DC-vaccines (see section 5).
Point 4. The recently published review and research articles should be cited and discussed in the revision, for example, Advanced Functional Materials 2019, 29 (49).
Following the Reviewer's suggestion we have added some recent reviews and original research articles, including Advanced Functional Materials 2019, 29 (49).
Reviewer 3 Report
This is a helpful, if poorly written review of the role of dendritic cells in cancer antigen presentation after immunogenic cell death. Authors are strongly advised to seek writing assistance since many sentences in their manuscript are sloppily written, contain incorrect syntax and therefore may mislead the reader. This also relates to utilization of certain incorrect terms, e.g. describing radiology and hyperthermia as 'physical therapy' (vs. accepted term of 'chemotherapy'). However, the definition of 'physical therapy' in English has nothing to do with cancer - it is 'a branch of rehabilitative health that uses specially designed exercises and equipment to help patients regain or improve their physical abilities.' Additionally, some of the papers cited in the review are a bit outdated and the approaches tested 10-15 years ago have been since mostly abandoned in favor of checkpoint inhibitors and CAR-T cell- based therapies, a fact, which is completely ignored by the authors. Notably, nearly all of research described in the manuscript is preclinical and has so far shown very little translational efficacy. Although authors do mention it briefly, this should discussed at a greater breadth. Finally, some sections of the manuscript lack internal logic and look like incoherent retelling of other scientists' results without any substantial analysis. Again, this kind of review is a helpful contribution to the literature, but should be drastically rewritten prior to publication.
Author Response
Comment #1. Authors are strongly advised to seek writing assistance since many sentences in their manuscript are sloppily written, contain incorrect syntax and therefore may mislead the reader.
Response Comment #1. Thank you for your comment. Our manuscript has been reviewed by a native English speaker and extensive changes have been made throughout all section (extensive changes are highlighted in yellow).
Additionally, in order to improve readability, Table 1 has been reformatted. Also, we are now including data from the following references:
- [Reference 149] Ji, J.; Fan, Z.; Zhou, F.; Wang, X.; Shi, L.; Zhang, H.; Wang, P.; Yang, D.; Zhang, L.; Chen, W.R., et al. Improvement of dc vaccine with ala-pdt induced immunogenic apoptotic cells for skin squamous cell carcinoma. Oncotarget 2015, 6, 17135-17146.
- [Reference 150] Garg, A.D.; Vandenberk, L.; Koks, C.; Verschuere, T.; Boon, L.; Van Gool, S.W.; Agostinis, P. Dendritic cell vaccines based on immunogenic cell death elicit danger signals and t cell-driven rejection of high-grade glioma. Sci Transl Med 2016, 8, 328ra327.
- [Reference 151] Zheng, Y.; Yin, G.; Le, V.; Zhang, A.; Chen, S.; Liang, X.; Liu, J. Photodynamic-therapy activates immune response by disrupting immunity homeostasis of tumor cells, which generates vaccine for cancer therapy. Int J Biol Sci 2016, 12, 120-132.
These studies were summarized in section 4.2.2, but, mistakenly, not included in the original table. We apologize for this mistake and appreciate the opportunity to correct it.
Comment #2. This also relates to utilization of certain incorrect terms, e.g. describing radiology and hyperthermia as 'physical therapy' (vs. accepted term of 'chemotherapy'). However, the definition of 'physical therapy' in English has nothing to do with cancer - it is 'a branch of rehabilitative health that uses specially designed exercises and equipment to help patients regain or improve their physical abilities.'
Response Comment #2. We thank the Reviewer for raising this point, and apologize for overlooking this potential source of confusion. Throughout the revised manuscript the term “physical therapy” has been replaced with “physical therapeutic modality”, in line with previously published work (see for instance Oncoimmunology. 2015 Jan 7;3(12):e968434. Physical modalities inducing immunogenic tumor cell death for cancer immunotherapy. Adkins I, Fucikova J, Garg AD, Agostinis P, Špíšek R.).
Comment #3. Additionally, some of the papers cited in the review are a bit outdated and the approaches tested 10-15 years ago have been since mostly abandoned in favor of checkpoint inhibitors and CAR-T cell- based therapies, a fact, which is completely ignored by the authors. Notably, nearly all of research described in the manuscript is preclinical and has so far shown very little translational efficacy. Although authors do mention it briefly, this should discussed at a greater breadth.
Response Comment #3. We thank the reviewer for this comment. Our review was aimed at outlining the current state of knowledge on preclinical/clinical development of anticancer DC vaccination using ICD-killed tumor cell lysates as antigenic and adjuvant cargo. We agree with the Reviewer that most of the studies conducted to date in this specific area of investigation have made use of preclinical models to test the prophylactic and/or therapeutic effectiveness of these vaccination protocols. Nevertheless, in this review we summarize the relevant findings of both preclinical and clinical studies, and discuss the obstacles, advantages and future perspectives for this specific type of DC-based vaccines in immuno-oncology.
Comment #4. Finally, some sections of the manuscript lack internal logic and look like incoherent retelling of other scientists' results without any substantial analysis.
Response Comment #4. We appreciate this criticism as it gives us the opportunity to improve our manuscript. We have revised all sections of the manuscript accordingly. In particular, we have expanded section 2 on “DC-based immunotherapy” and the discussion section.
Round 2
Reviewer 3 Report
Much improved over the previous version (language, presentation, analysis, terminology). Suitable for publication.